# Maternal Cancer and Perception of Child Psychological Adjustment: The Role of Mother’s Anxiety, Depression, and Parenting Stress

**DOI:** 10.3390/cancers15030910

**Published:** 2023-01-31

**Authors:** Alessandra Babore, Carmen Trumello, Tânia Brandão, Alessandra Cavallo, Sonia Monique Bramanti

**Affiliations:** 1Department of Psychological, Health and Territorial Sciences, School of Medicine and Health Sciences, University «G. d’Annunzio», Via dei Vestini, 31, 66100 Chieti, Italy; 2Williams James Center for Research, ISPA—Instituto Universitário, 1149-041 Lisbon, Portugal; 3Center for Psychology at the University of Porto, 4200-135 Porto, Portugal

**Keywords:** cancer, mother, parenting stress, anxiety, depression, child adjustment

## Abstract

**Simple Summary:**

In the last decades, the incidence of early-onset cancers, defined as cancers diagnosed in adults <50 years of age, has been rising in many countries. A diagnosis of cancer in a mother is a burden not only for the patient herself but also for the entire family system and particularly for her minor children. Despite its relevance, this topic has been little explored in the existing literature. The aim of our study was to analyze aspects associated with children’s psychosocial adjustment regarding maternal cancer. Following this general objective, we considered age and sex of the child and factors related to maternal cancer and mothers’ psychological adjustment. We found that the time passed since the diagnosis and the levels of mothers’ anxiety and parenting stress were associated with children’s difficulties. These findings may help health professionals to develop structured and tailored programs to support both mothers and children after an oncological diagnosis.

**Abstract:**

A mother’s cancer diagnosis may have consequences for all family members, particularly for children, since it makes a parent less physically and emotionally available, with effects on the relationship with the child and his/her development. The main aim of this study was to analyze children’s psychological adjustment in the context of maternal cancer by considering factors related to the child (i.e., age and sex), the mother (psychological adjustment), and cancer (i.e., time elapsed from the diagnosis and current oncological treatment). Self-report questionnaires investigating mothers’ parenting stress, depression, anxiety, and children’s emotional and behavioral problems were administered to 124 mothers (mean age = 44.52 years; SD = 7.22) diagnosed with cancer. A hierarchical regression analysis highlighted that time since diagnosis and mothers’ anxiety and parenting stress accounted for almost 44% of the variance of the children’s difficulties. Maternal depression, current oncological treatment, and the child’s age and sex were not statistically significant. Higher mothers’ parenting stress and anxiety and a longer time elapsed since the first diagnosis predicted higher levels of children’s emotional and behavioral problems. These findings highlight the need to identify precursors of childhood distress in the context of maternal cancer and to develop structured programs to support both mothers and children.

## 1. Introduction

Cancer in a parent is a burden not only for the patient but also for the entire family system [1]. A stressful life event such as parental cancer may affect children’s development. Compared to normative groups, the children of ill parents have more emotional and behavioral problems [2], probably because they depend on the care and support of their caregivers, and this makes them particularly vulnerable to parental cancer [2,3].

Over the last decades, the incidence of early-onset cancers, namely, cancers diagnosed in adults <50 years of age has increased in several countries, with a large number of parent-age patients [4]. About 24% of cancer patients have minor children [5]; when cancer occurs in a parent, the diagnosis and its treatment make him/her less available physically and emotionally [6,7] and cause a disruption of the daily routine and changes in the family structure and functioning [5,8]. This stressful event and its consequences can affect children’s healthy development, which is optimized when parents are responsive, supportive, and sensitive to their needs [9,10].

Previous studies reported that many factors may influence children’s development when a parent has cancer; these factors are related to the disease type and to child’s and parent’s characteristics [1]. Among the child-related factors reported in the existing literature, there are age and sex, but the results from research are not univocal; in fact, many studies observed that latency-age (6–10 years) sons have more emotional and behavioral problems than same-age daughters [2,11,12]; other studies reported more difficulties in adolescent daughters than in their male counterparts [13,14,15].

Negative outcomes in children may depend also on factors related to illness. A recent diagnosis and/or being under treatment could make the mother less available and increase a child’s difficulties. Among the medical variables that could affect the child’s development there are advanced-stage disease, long disease duration, poor prognosis, the time elapsed from the diagnosis, recurrent illness, mediatized cancer, extensive surgery, radiotherapy and chemotherapy side effects [3,14,16]; however, the results are not consistent, in fact some studies have not identified any association between the objective medical variables and the child outcomes [1,17].

Some studies highlighted that negative affects in the parent are associated with negative parent–child relationships and a worse psychological functioning in the child [3,18]. Parents with cancer experience more negative affective states such as depressed mood, anxiety symptoms, and stress [3,13,19], which make them less available and may influence the parent–child relation [20], with consequences on the emotional and behavioral children’s development [13,17]. As for parental psychological maladjustment in this context, parenting stress is a specific type of stress that depends on the disparity between the demands of being a parent and the resources perceived as available to cope with those demands [21,22]. When parenting stress arises, it might impact both parents and children, leading to maladaptive parenting behaviors and affecting the parent–child relationship. According to the Abidin’s model [21], parenting stress may depend on parental factors, such as anxious and depressive symptoms, child factors, such as demandingness and temperament, and situational factors, such as perceived social support or role restriction. Previous studies among families without an oncological diagnosis found that parental stress could negatively affect the parents’ attitudes towards the children and their perception of a child’s emotional and behavioral difficulties [23,24]. To the best of our knowledge, no previous study has explored parenting stress in the context of a maternal oncological diagnosis.

The current study aimed to fill this gap, analyzing mothers’ perception of children’s adjustment in the context of maternal cancer. More specifically, we explored if and to what extent factors related to the child (i.e., child’s age and sex), to the oncological diagnosis (i.e., time elapsed from diagnosis and current oncological treatment), to mothers’ individual adjustment (i.e., levels of depression and anxiety), and to mothers’ relational adjustment (i.e., parenting stress) may be associated with a child’s psychological problems, as perceived by the mothers.

## 2. Materials and Methods

### 2.1. Participants and Procedure

The study involved 124 women, with a mean age of 44.52 years (SD = 7.22). They were recruited online and from some oncology units located in middle-sized cities of central and southern Italy. The inclusion criteria comprised having had a cancer diagnosis in the lifetime and having a child aged 0–17 years. When mothers had more than one child (60% of the recruited sample), we asked them to refer to the youngest child in their answers to the questionnaires.

A letter containing detailed information on the main study aims and rationale was given to all participants. The participating women were informed that their participation was voluntary and that their responses would be confidential. Written informed consent was collected from every participant. We alphanumerically coded the questionnaires to guarantee anonymity.

All the employed procedures and measures used in the current study were fully compliant with the Declaration of Helsinki and with the Ethics Code of the Italian Board of Psychology—the regulatory Authority that provides the national guidelines for research and clinical practice—and the study was approved by the Institutional Review Board of the Department of Psychological, Health and Territorial Sciences [Protocol Number: 21001].

### 2.2. Measures

#### 2.2.1. Socio-Demographic and Medical Characteristics

We developed a socio-demographic and medical questionnaire to assess general information about the enrolled women (e.g., age, education level, employment, and marital status), their children’s characteristics (e.g., sex and age), and general information about their cancer diagnosis (e.g., time since diagnosis and cancer treatment).

#### 2.2.2. Strengths and Difficulties Questionnaire (SDQ)

To measure the maternal perception of children’s emotional and behavioral problems, the Strengths and Difficulties Questionnaire (SDQ; [25]; Italian version: [26]) was administered to the mothers. It is a 25-item screening instrument (sample items: My child is… “restless, overactive, cannot stay still for long”), rated on a 3-point Likert scale (not true, somewhat true, certainly true). The SDQ is composed of five subscales that assess the difficulties and resources of the child; the prosocial subscale measures a positive aspect of behavior; the other four subscales (emotional problems, conduct problems, hyperactivity, peer problems) reflect the difficulties. In the present study, we considered the total difficulties score, obtained from the sum of these last four subscales. The reliability of this instrument in this study was satisfactory (Cronbach’s alpha = 0.77).

#### 2.2.3. The Parenting Stress Index-Short Form (PSI-SF)

The Parenting Stress Index-Short Form (PSI-SF; [21]; Italian version: [27]) is a 36-item measure completed by the mothers to assess their parental stress. Each item (e.g., “My child rarely does things for me that make me feel good”) is rated on a 5-point Likert scale ranging from 1 (strongly disagree) to 5 (strongly agree). The PSI-SF includes a total stress score, that indicates the level of stress a person is feeling in his/her parental role, and three subscale scores, each composed of 12 items: parental distress, that measures the parents’ perception of their behavior; parent–child dysfunctional interaction, that measures the parents’ view of expectations and interactions with their child; and difficult child characteristics, that measures the parents’ perceptions of their child’s temperament. In the present study, we considered the total stress scores, obtained from the sum of the three subscales. The reliability of this instrument in the present study was good (Cronbach’s alpha = 0.92).

#### 2.2.4. The Zung Depression Self-Rating Scale (SDS)

To investigate the depressive symptoms, the Zung Depression Scale (SDS; [28]; Italian version: [29]) was used. It consists of 20 items (e.g., “I feel that others would be better off if I were dead”) that evaluate the affective, psychological, and somatic symptoms of depression. Each item is rated on a 4-point Likert scale that specifies how frequently the symptom is experienced (ranging from 1 = some of the time to 4 = most of the time). Higher scores are associated with higher depression. The reliability of this instrument in the present study was satisfactory (Cronbach’s alpha = 0.84).

#### 2.2.5. The Zung Anxiety Self-Rating Scale (SAS)

The Zung Anxiety Scale (SAS; [30]) is a self-report questionnaire composed of 20 items (e.g., “I feel more nervous and anxious than usual”) designed to assess anxious symptoms, both affective and somatic. Each item is rated on a 4-point Likert scale ranging from 1 (some of the time) to 4 (most of the time); the total score ranges from 20 to 80, with higher scores signaling higher levels of anxious symptomatology. The reliability of this instrument in the present study was good (Cronbach’s alpha = 0.82).

#### 2.2.6. Statistical Analyses

Statistical analyses were performed with the Statistical Package for the Social Sciences (SPSS, Version 19, Chicago, IL, USA) [31]. First, we tested the skewness and kurtosis of all the variables to check their distribution. Descriptive statistics were used to investigate the participants’ sociodemographic and medical characteristics. 

Hierarchical regression analysis was carried out to identify the independent correlates of child’s difficulties (evaluated by the SDQ). The first level included children’s sociodemographic characteristics (age and sex). On the second level, illness-related factors (time elapsed since diagnosis and current cancer treatment) were entered. Current cancer treatment was evaluated by means of a dummy variable (with 1 = under chemotherapy or radiation therapy, and 0 = not under treatment). The third level included the mothers’ depressive and anxious symptoms. The last level comprised maternal parenting stress. The significance level α was set at 0.05.

## 3. Results

The sample comprised 124 women with a cancer diagnosis. All participants were parents of children younger than 18 years. The characteristics of the participants are presented in Table 1. 

The children had a mean age of 10.68 (SD = 5.82) years and were equally distributed by sex (50% females). The skewness and kurtosis of all the variables were between −1 and +1; thus, we considered their distributions as acceptably normal [32]. 

As for the aim of exploring factors associated with children’s psychological problems, the results of the multivariate regression analysis are synthetized in Table 2.

The children’s sociodemographic characteristics (age and sex) accounted for less than 1% of the variance of the children’s difficulties (*p* = 0.612). Adding in the second step the time elapsed since the diagnosis and the presence or not of a current oncological treatment, the model accounted for an additional 4.7% of the explained variance (*p* = 0.147). In the third step, maternal anxiety and depression were added, but only the former significantly contributed to child’s difficulties, with 29.6% of the explained variance (*p* = 0.000). In the final step, maternal parenting stress added a 14.3% to the explained variance (*p* = 0.000). 

Overall, in the final model, higher mothers’ parenting stress and anxiety and longer time elapsed since the first diagnosis predicted higher children’s problems. Maternal depression, current oncological treatment, and child’s age and sex were not statistically significant.

## 4. Discussion

In the current study, we aimed to examine associative factors of children’s psychosocial adjustment with regard to maternal cancer. Following this general objective, we considered children’s characteristics (namely, age and sex), mothers’ illness-related factors (time since diagnosis and current oncological treatment), and mothers’ psychological adjustment at an individual (depression and anxiety) and a relational (maternal parenting stress) level. To our knowledge, the associations between the perception of children’s difficulties, maternal adjustment, and mothers’ parenting stress have been little explored in children of mothers diagnosed with cancer [17,18].

Our findings showed that, as regard the characteristics of the children, age and sex were not significantly related to their difficulties. These results are partially in agreement with the existing literature, which produced inconsistent findings for both age and sex. Following the review of Krattenmacher et al. [33], studies with lower methodological quality found associations between children’s age and adjustment, while others with middle to high rated quality did not. As for the gender differences, some studies reported more emotional and behavioral problems in daughters than in sons, and others found no significant gender effects. However, the studies detecting significant gender differences were judged as characterized by low and middle methodological quality [33]. Overall, we can conclude that our findings are consistent with previous literature with high methodological quality. However, more research is needed to further analyze this topic. For instance, it might be hypothesized that daughters, with respect to sons, could be more concerned with specific types of cancer that involve female aspects of the body (such as breasts, ovaries, and uterus) [34]. This could be an interesting issue to explore in subsequent research.

As for the illness and treatment characteristics, the time elapsed from the diagnosis resulted to be slightly associated with children’s difficulties. Our results show that the longer the time elapsed since the diagnosis, the greater the child’s difficulties. This finding is quite surprising, as it might be expected that a longer time from the diagnosis may reduce the impact of the disease on the child’s adjustment. A possible explanation may be related to the fact that the diagnosis represents a period of significant distress for all family members who need time to react and adjust to this new problem. Therefore, psychological difficulties can emerge only after a certain period from the diagnosis. In line with this hypothesis, a study carried out in families of children with cancer highlighted that the time since the diagnosis positively moderated the effect of parental depression on child’s anxiety, so that parents’ and children’s symptoms were more strongly associated in families with a greater amount of time elapsed since the diagnosis [35]. It could also be hypothesized that an oncological diagnosis is an ongoing problem [36] with long-term outcomes [37] that affect not only the patient but also the whole family [3]. However, we may not exclude that other factors not explored in the present research may contribute to this finding. Hence, further studies, with a prospective design, are needed to deepen this topic, also considering, for example, the duration of the active disease, that a previous study found as a risk factor for child development [38].

As regards maternal psychological functioning, parenting stress was highly associated with child’s functioning. Some studies have investigated which variables influence the development of a child in the presence of a parental cancer [1,12], but to our knowledge, this is the first study to identify parenting stress as a possible predictor of a child’s difficulties when a parent has a cancer diagnosis. Few studies in other contexts found that a high parenting stress is an important risk variable for children development, with consequences on the child’s socio-emotional development [39] and increased behavioral problems [40,41]. Classically, parenting stress has been defined as a parental negative experience resulting from a disparity between the perceived demands about parenting and the available resources [10,21]. We may hypothesize that a cancer diagnosis may decrease mothers’ parental resources at different levels: in terms of time, as medical examinations and treatments can take a long time, reducing mothers’ availability for the children; in terms of physical strength, since some treatments can negatively affect their physical well-being and, consequently, their ability to respond to the children’s requests; in terms of emotional availability, as a cancer diagnosis may lead mothers to focus on their illness and subsequently be less responsive and sensitive to their offspring’s needs. According to previous research [42,43], mothers who experience high parenting stress may be less supportive during distressing situations and may perceive their children’s and/or their parental functions as very challenging. These negative perceptions and feelings may hinder their ability to buffer their children’s stress and to teach them how to manage their emotions and promote adaptive behaviors. Hence, these maternal processes and behaviors could affect the quality of their interactions with the children, making the latter more prone to emotional and behavioral negative outcomes.

Furthermore, we found that anxious symptoms in mothers were associated with children’s difficulties. These data are consistent with previous studies detecting a relation between parental anxiety and children’s adjustment and development [44]. In a large community sample, Clavarino et al. [45] found that maternal anxiety was strongly associated with children’s attention problems at 5 and 14 years. In our findings, we did not detect any association between maternal depressive symptoms and children’s difficulties, a result not in line with some existing research that found that depressive symptoms and general family functioning were factors that could influence the development of a child [12,13,46,47]. However, some previous studies found a pattern similar to ours, highlighting that parents’ anxiety but not depression predicted higher levels of child’s symptoms [48,49]. A possible hypothesis to explain this result might be that parents with higher levels of anxiety may misinterpret child-related information and, consequently, be less supportive and emotionally available, which would lead to the development of greater difficulties in their children [42,50]. An alternative explanation could be based on methodological issues of our research: since we used self-report questionnaires filled in only by the mothers, it cannot be excluded that more anxious mothers attributed higher levels of emotional problems to their children. During distressing situations, such as those represented by dealing with cancer, mothers’ anxiety and worries may have had an impact on their perception of the children’s requests and behaviors. However, further studies with multi-informant procedures should better explore this issue.

Despite the contribution of this study, it is important to highlight its limitations that should be addressed in future research. Firstly, due to the correlational nature of the study, no causal relationships among the considered variables may be inferred; indeed, we cannot exclude that the children’s behavioral and emotional difficulties may affect mothers’ parenting stress. Future longitudinal studies are important to assess causality among the variables and specially to explore and better understand the role of time since cancer diagnosis on child’s development. Secondly, as abovementioned, the collected data could be biased by a self-selection and a one-informant procedure; in fact, only the mothers filled in the questionnaires regarding their psychological adjustment and children’s emotional and behavioral difficulties; hence, it cannot be excluded that the levels of the mothers’ distress, associated with the cancer-related health condition, may have caused a higher perception of distress in the children that could not correspond to their actual adjustment. Thirdly, this study did not consider the paternal contribution to the children’s outcomes. Fourthly, some important medical variables (i.e., duration of the illness and cancer stage) were not examined in the present research. Lastly, as a control group of cancer-free mothers is missing, we cannot quantify the actual impact of cancer on our findings. All these limitations prevent us from generalizing the results.

## 5. Conclusions

Our research is the first to explore associations between children’s development and parental stress and adjustment in a group of mothers diagnosed with cancer. A cancer diagnosis in a mother may have consequences on all family members and particularly on children, since it makes a parent less physically and emotionally available, with effects on the relationship with the child and on his/her development [5,7]. This finding highlights the need to identify precursors of childhood distress in the context of a parental oncological disease and to develop structured programs to support both parents and children for a long time after the diagnosis. Therefore, the object of psycho-oncological care should be broadened from the patient to the whole family system, including the offspring, particularly if these are children and adolescents.

Most previous studies investigated the psychological effects of children cancer on mothers, but few previous studies focused on the impact of maternal cancer on her parental role and on children’s development. This lack is surprising, especially when we consider the increased incidence of cancer in young women in the last decades [51]. An important step to improve the psychosocial care of these women could be to explore the role of parenting concerns in a patient’s process of adaptation to the disease [52]. Mothers with cancer should be supported by mental health professionals in reducing individual distress and parenting stress, through specific counseling services. As parenting stress is a considered a common experience for parents across all contexts and sociodemographic groups [53], it should be a priority research topic and an area of clinical interest, especially in at-risk populations. The current study tried to address this issue, but further research on this crucial topic is needed.

## Figures and Tables

**Table 1 cancers-15-00910-t001:** Demographic and clinical characteristics of the study participants (N = 124).

Participants’ Characteristics	(N = 124)
Age (M ± SD)	44.52 ± 7.22
Education	
Middle school or less	20
High school or equivalent	55
College degree or higher	49
Occupation	
Employed	92
Housewife	32
Diagnosis	
Breast cancer	76
Gynecological cancer	14
Other Types	34
Time from diagnosis (in years?)	6.02 ± 7.19
Under chemotherapy/radiotherapy treatment	
yes	31
no	93
Marital status	
Relationship	107
No relationship	17
Marital relationship duration (in years)	13.57 ± 8.71
Children	
N = 1	49
N ≥ 2	75
Child’s Sex	
male	62
female	62
Age of Children (in years)	10.68 ± 5.82
Children’s Difficulties (SDQ)	7.81 ± 4.85
Parenting Stress (PSI-SF)	79.60 ± 21.70
Anxiety (SAS)	37.35 ± 7.80
Depression (SDS)	39.19 ± 8.91

*Note.* The content was generated by the authors. Abbreviations: M = mean; SD = standard deviation; SDQ = Strengths and Difficulties Questionnaire; PSI-SF = Parenting Stress Index-Short Form; SAS = Zung Anxiety Self-Rating Scale; SDS = Zung Depression Self-Rating Scale.

**Table 2 cancers-15-00910-t002:** Regression analysis for children’s difficulties (dependent variable).

Variables	R^2^	ΔR^2^	B	β	SE	t
**Step 1**	0.008					
Children’s age			−0.54	−0.065	0.075	0.721
Children’s sex			−0.614	−0.064	0.874	0.703
**Step 2**	0.055	0.047				
Children’s age			−0.070	−0.074	0.074	−0.935
Children’s sex			−0.569	−0.059	0.861	−0.661
Under treatment			−0.098	−0.009	1.042	−0.095
Time since diagnosis			0.145	0.215 *	0.063	2.292
**Step 3**	0.296 ***	0.241				
Children’s age			−0.055	−0.066	0.065	−0.849
Children’s sex			−0.373	−0.039	0.759	−0.491
Under treatment			0.982	0.088	0.923	1.064
Time since diagnosis			0.183	0.271 **	0.055	3.296
Anxiety			0.205	0.331 **	0.067	3.052
Depression			0.115	0.212	0.059	1.947
**Step 4**	0.439 ***	0.143				
Children’s age			−0.089	−0.107	0.059	−1.523
Children’s sex			−1.028	−0.107	0.691	−1.488
Under treatment			0.706	0.063	0.829	0.851
Time since diagnosis			0.114	0.169 *	0.051	2.221
Anxiety			0.172	0.277 **	0.061	2.837
Depression			−0.010	−0.018	0.058	−0.174
Parenting Stress			0.105	0.471 ***	0.019	5.430

* *p* < 0.05; ** *p* < 0.001; *** *p* < 0.001. *Note.* The content was generated by the authors. Abbreviations: SE = standard error.

## Data Availability

The data presented in this study are available on request from the corresponding author. The data are not publicly available due to privacy restrictions.

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
