# Peer review of "Maternal Cancer and Perception of Child Psychological Adjustment: The Role of Mother’s Anxiety, Depression, and Parenting Stress"

_cancers, 2023, doi:10.3390/cancers15030910_

Round 1

Reviewer 1 Report

Manuscript number: cancers-2123693

Title:  Maternal cancer and child psychological adjustment: The role of
mother’s anxiety, depression, and parenting stress

Thank you for the opportunity to review this work on investigating factors associated with maternal cancer that affected children’s emotional or behavioral difficulties. The study is clinically relevant, but I do have some questions and comments that need to be addressed by the authors. 

1.    Since parenting stress seemed to be a major finding from this study, this aspect may deserve some more explanations about how the hypothesis that parental stress may be related to children’s difficulties was formed. For example, although the authors have mentioned that past studies highlighted that negative affects in parents, such as depression or anxiety, may have influenced children’s development, no further literature between parenting stress, including parent-child dysfunctional interactions or the perceptions of child's temperament, and children’ difficulties have been summarized. Please provide the point.

2.    Is there any way to know whether the influences of parenting stress on children’s difficulty was related to cancer? Because it is also possible that even in the absence of cancer, parenting stress and children's difficulties may still be positively correlated. For example, if the parenting stress was greater, the child may have more behavioral problems; or if the child has more behavioral or emotional problems, the parenting stress were greater. How do the authors take this into consideration?

3.    As above, the way to find out may require a comparison group of parents without cancer. If not possible, the authors might need to recognize this as a limitation.

4.    Again, it seems that parenting stress is an important factor, but relevant mechanisms of how this may be related to children’s difficulties, or probably associations between children’s difficulties and different items from the questionnaire may be described or explained more in the Discussion section. Currently only the predicting role of parenting stress was emphasized but not possible mechanisms that readers may be interested to know.

5.    Reasons of negative outcomes should be discussed, for example, why didn’t maternal depression have any effects on children’s difficulties, as opposed to anxiety symptoms? Comparisons with past literature and possible mechanisms for the results in this study should be addressed.  

6.    If parenting stress and other factors important predictors of children’s difficulties, what kind of interventions may be helpful? This may be added to give the readers a direction of clinical, research, or public health implications.

Reviewer 2 Report

Comments and Suggestions for Authors

The article “Maternal cancer and child psychological adjustment: The role of mother’s anxiety, depression, and parenting stress” treats a relevant and underexplored topic. While the previous literature abounds in studies that have examined the effect of a child's cancer diagnosis on parents, this is one of the few studies that addresses the associated psychological difficulties among children whose mothers have cancer. The study brings a novelty by showing that parental stress is significantly associated with child’s difficulties when a parent has cancer, negatively affecting the socio-emotional development of the latter.

The article has a logical, coherent structure, easy for the reader to follow, with documented points of view selected from 46 bibliographic sources, relevant to the studied topic, of which 5 are from the last five years. The authors used with moderation the self-citations.

Each section was clearly described and comprehensive. The methodology used is adequate for the aim of the study. The research design is pertinent for testing the hypotheses. The sampling procedure is sufficiently explained and appropriate. Ethics of research issues are adequately described in the manuscript. It is appreciable the way in which the authors have identified and exposed the limitations of the research.

In order to publish the article, I recommend the following little adjustments:

- In the paragraph comprised between the lines 168-179, please indicate which of the mentioned values are not shown in the Table 2.

- Under the Tables 1 and 2, please insert a note specifying whether the content is generated by the authors. If not, please indicate the source of data.

- Please translate into English the Italian title of the article specified at the 27th bibliographic source from the References section.

In conclusion, I recommend publishing the article with minor revisions because it is of high interest not only by researchers, but also for the large public.

Submission Date

09 January 2023

Reviewer 3 Report

Review of : 'Maternal cancer and child psychological adjustment: The role of mother's anxiety, depression, and parenting stress'

Dear Editor,

This study deals with a very sensitive and important topic, which may contribute to explaining the psychological factors involved in the diagnosis of cancer in mothers, with respect to the perception of maladjustment in their children.

The article summarises the existing literature very well, the introductory arguments are clear and the bibliography is up-to-date.

The statistical methods used are adequate and the tables clearly show the results.

However, there are inconsistencies of a methodological nature, which make the aspect of measuring psychological adjustment unclear. Evidence:

- Title : "child psychological adjustment"

- Abstract line 24 " mother (psychological adjustment) "

- Introduction line 75 "The current study aimed at analysing children's adjustment".

- Method lines 106-107 " To measure children's emotional and behavioral problems, the Strengths and Difficulties Questionnaire (SDQ; [23]; Italian version: [24]) was administered to mothers."

- Discussion line 186 : "To our knowledge, the associations between children's difficulties, maternal adjustment and mothers' parenting stress have been little explored in children [...]".

These points mistakenly lead to believe that psychological adjustment has been measured in children; whereas, the method highlights that psychological adjustment of children has not been measured in children, but in mothers' perceptions.

This problem of clarity must be resolved before the study can be considered for publication. The title may erroneously suggest that two samples were examined: mothers and children.

Even in the discussion and in the conclusions this ambiguity persists. Therefore, the main reason for concern and clarification is to underline that the sample consists of mothers only, ( the study was done on mothers, mothers filled in the questionnaires, mothers expressed their perceptions of the child's distress) .

I think this can make the work more reliable, accurate and precise.

Furthermore, the fact that the psychological adjustment of the children was measured by the mothers could be a limiting factor. It has not been estimated whether the effect of depression of anxiety or parenting stress, related to the cancer-related health condition, may have caused the mothers to perceive greater distress in their children ( as an expression of their own concern and mental state ). Statistical estimates of these effects are lacking.

Once the methodological focus and ambiguities regarding the observed sample have been clarified, the study is certainly re-evaluable for publication because it raises a very important and neglected issue.

Round 2

Reviewer 1 Report

Thank you for the revision. There's only one comment from me this time:

Line 82, is there some grammar issues with the sentence of 'To the best of our knowledge, any previous study explored parenting stress in the context of maternal oncological diagnosis.' Does the author really want to mean that 'no study has ever explored parenting stress? If so, the sentence should be revised to : 'To the best of our knowledge, no previous study has explored parenting stress in the context of maternal oncological diagnosis.' 

Author Response

Thank you for the revision. There's only one comment from me this time:

Line 82, is there some grammar issues with the sentence of 'To the best of our knowledge, any previous study explored parenting stress in the context of maternal oncological diagnosis.' Does the author really want to mean that 'no study has ever explored parenting stress? If so, the sentence should be revised to : 'To the best of our knowledge, no previous study has explored parenting stress in the context of maternal oncological diagnosis.' 

We want to thank the Reviewer for his/her careful reading of our manuscript. We have made the requested change in line 82.